# Identifying therapeutic targets for cancer among 2074 circulating proteins and risk of nine cancers

Karl Smith-Byrne [1] ✉, Åsa Hedman[2,3], Marios Dimitriou [2,3], Trishna Desai[1], Alexandr V. Sokolov[4], Helgi B. Schioth[4], Mine Koprulu [5], Maik Pietzner [5,6,7], Claudia Langenberg [5,6,7], Joshua Atkins [1], Ricardo Cortez Penha[8], James McKay[8], Paul Brennan[8], Sirui Zhou [9], Brent J. Richards [10], James Yarmolinsky[11], Richard M. Martin [11,12,13], Joana Borlido [14], Xinmeng J. Mu[15], Adam Butterworth [16], Xia Shen [17], Jim Wilson[17], Themistocles L. Assimes [18], Rayjean J. Hung [19], Christopher Amos [20], Mark Purdue[21], Nathaniel Rothman[22], Stephen Chanock [21], Ruth C. Travis[1,23], Mattias Johansson[8,23] & Anders Mälarstig[2,3,23]

Circulating proteins can reveal key pathways to cancer and identify therapeutic targets for cancer prevention. We investigate 2,074 circulating proteins and risk of nine common cancers (bladder, breast, endometrium, head and neck, lung, ovary, pancreas, kidney, and malignant non-melanoma) using cis protein Mendelian randomisation and colocalization. We conduct additional analyses to identify adverse side-effects of altering risk proteins and map cancer risk proteins to drug targets. Here we find 40 proteins associated with common cancers, such as *PLAUR* and risk of breast cancer [odds ratio per standard deviation increment: 2.27, 1.88-2.74], and with high-mortality cancers, such as *CTRB1* and pancreatic cancer [0.79, 0.73-0.85]. We also identify potential adverse effects of protein-altering interventions to reduce cancer risk, such as hypertension. Additionally, we report 18 proteins associated with cancer risk that map to existing drugs and 15 that are not currently under clinical investigation. In sum, we identify protein-cancer links that improve our understanding of cancer aetiology. We also demonstrate that the wider consequence of any protein-altering intervention on well-being and morbidity is required to interpret any utility of proteins as potential future targets for therapeutic prevention.

Proteins govern cellular action in all human biological processes and are crucial for our defences against both the onset and progression of cancer. Identifying circulating proteins important to the aetiology of cancer may improve our understanding of pathways leading to cancer and highlight potential targets for therapeutic prevention. Circulating proteins are valuable candidate targets for drug development since drug-target engagement can be evaluated in the bloodstream during randomised control trials (RCTs), accelerating target development. Additionally, identifying circulating cancer risk proteins allows for the subsequent selection of future RCT participants with risk-inducing protein concentrations, which may improve RCT effectiveness. Developing therapeutic prevention strategies, either alone or as a

complement to existing prevention programs, such as smoking cessation, are urgently needed given cancer burden is projected to double by the year 2040[1].

Therapeutic prevention is an effective and commonly used strategy for the primary prevention of some common chronic diseases. Prevention strategies have thus far been most successful for cardiovascular disease using statins that target the HMG-CoA reductase protein as a first-line treatment to lower low-density lipoprotein (LDL) cholesterol[2,3]. In contrast, efforts to identify targets for the therapeutic prevention of cancer have been less fruitful, hampered by a more complex aetiology and difficulty identifying potential targetable aetiological biomarkers[4]. Exceptions include therapeutically targeting the oestrogen receptor (ER) to prevent breast cancer[5] and *COX2* to prevent colorectal cancer in high-risk individuals[6]. Additional aetiological proteins for specific cancers have been identified, such as the role of higher levels of insulin-like growth factor-I in the development of breast[7], colorectal[8], and prostate[9,10] cancers, and of higher microseminoprotein-beta with lower prostate cancer risk[11]. Together these examples highlight the opportunity that may result if aetiological proteins for cancer are identified and the feasibility of using these to develop therapeutic prevention tools where high-risk populations are well-defined.

Identifying candidate aetiological biomarkers for cancer risk has traditionally involved analysing specific hypothesis-driven markers for single cancer outcomes in pre-diagnostic blood samples and comparable controls from large prospective cohorts[9,12,13]. The advent of high-throughput platforms that can measure hundreds to thousands of biomarkers simultaneously using small sample volumes has enabled hypothesis-free discovery analyses, but costs remain prohibitively high. An alternative cost-effective approach, that also limits bias by confounding and reverse causation, is to use robust genetic proxies of blood biomarkers to evaluate their aetiological relevance along the lines of Mendelian randomisation (MR)[14,15]. Using such MR-based approaches facilitates simultaneously querying thousands of markers in relation to the risk of multiple cancers using genome-wide

association data, which can identify risk markers and assess their association with one or multiple cancers. Proteins represent a particularly appealing application of MR as the blood concentrations of many proteins are regulated by genetic variants, many of which lie in or near a protein's cognate gene (variants known as *cis* protein quantitative trait loci [cis-pQTL])[16]. Cis-pQTL likely influence biological processes directly, such as by transcription or translation, making them less prone to common sources of bias in MR studies like horizontal pleiotropy[17]. It is also possible to complement cis-pQTL-based MR analyses with colocalisation analyses to further exclude confounding by linkage disequilibrium[18]. These methodologies allow for the *in-silico* simultaneous evaluation of the role of thousands of proteins in the aetiology of common cancers with high specificity.

In the current study, we estimated the associations of 2074 circulating proteins with the risk of nine common cancers using data from a total of 337,822 cancer cases. We aimed to identify cancer-risk proteins and assess whether these proteins may cause multiple or specific cancers. Where possible, we mapped risk proteins to potential therapeutic interventions and used MR and colocalisation phenome-wide association analyses (PHEWAS) to describe the promise and complexities that may result from intervening on risk proteins in terms of potential adverse outcomes.

## Results
### Protein effects on cancer risk
In total, 4507 of the 4698 cis-pQTL were available for analysis with at least one cancer site [min: 3308 cis-pQTL for bladder cancer and max: 4303 cis-pQTL for endometrial cancer], which represented 2023 of 2074 proteins with cis-pQTL included in our study [min: 1692 for bladder cancer and max: 1934 proteins for skin cancer] (Fig. 1., Supplementary data 2). MR and colocalization analyses identified 40 proteins (Supplementary data 3, Fig. 2.) with an association with at least one cancer site [min: one protein for ovarian cancer, max: 21 proteins for breast cancer]. A further 428 proteins were identified to

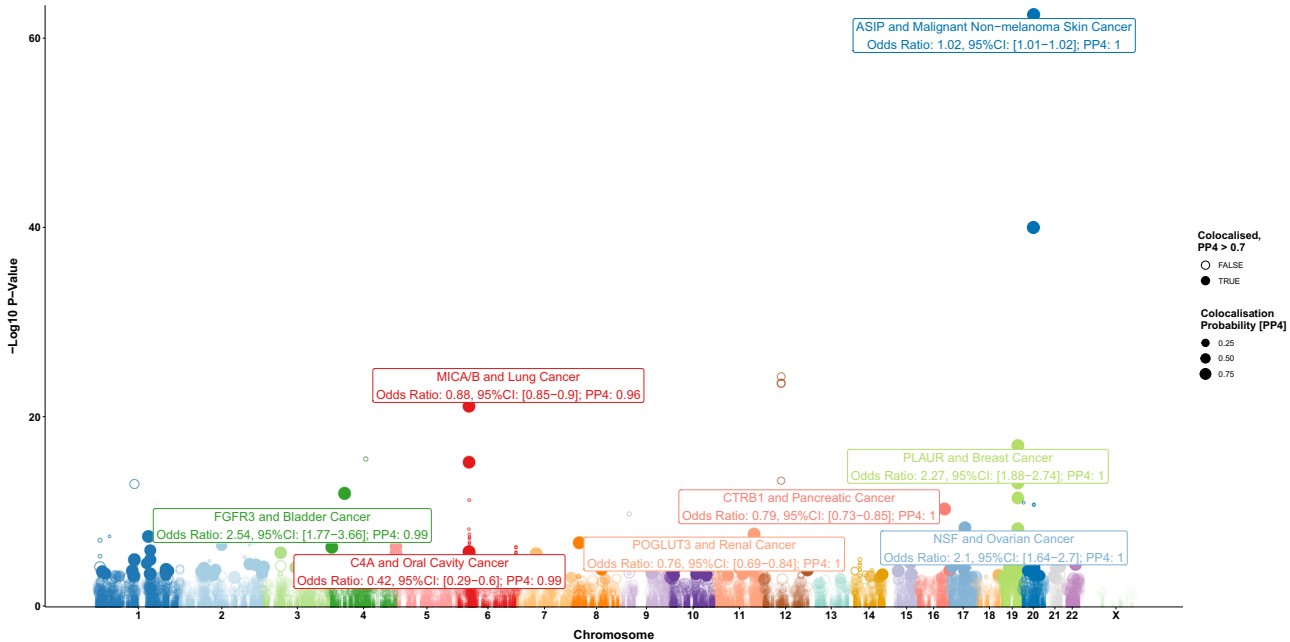

**Fig. 1 | Manhattan Plot for the association of genetically predicted protein concentrations with cancer risk.** Association of genetically predicted protein concentrations with cancer risk estimated using Wald ratios presented as a Manhattan plot where position is given by *cis-pQTL* coordinate with a selection of cancer risk associations additionally labelled for their association with cancer risk and colocalization probability (PP4). Top result for each cancer endpoint provided.

All tests are two-sided. Points highlighted as filled-in are those with PP4 > 0.7 with point size reflecting PP4 magnitude, which can vary between 0 and 1. Risk associations with MR *p* > 0.05 were not subject to colocalization analyses. Results labelled are those passing correction for multiple testing for the number of proteins analysed in this study per-cancer.

have evidence of colocalization [PP4 > 0.7] and at least a nominally significant MR association with the risk of cancer [min: 8 proteins for skin cancer and max: 241 for breast cancer] (Fig. 1, Supplementary data 2). We observed limited evidence for the association of proteins with risk for clear cell ovarian cancer, ever smoking lung cancers, HER2 enriched, luminal B, and luminal B-HER2 negative breast cancers. Additionally, we did not identify any proteins as a risk factor for cancer from multiple, independent cis-pQTL in MR analyses. Results by cancer site are summarised below. We did not find supportive evidence after correction for multiple tests for the association of cancer risk with protein levels for any protein identified in our main risk analyses (Supplementary data 3).

### Hormone-related cancers

**Breast.** We found 21 proteins associated with breast cancer risk. Among these proteins, nine (*AOC2, SPN1, CD160, RALB, GDI2, CPNE1, ULK3, CTSF, PLAUR*) were colocalised and had at least nominal associations with multiple molecular subtypes of breast cancer. *PLAUR* was observed to have a strong positive association with risk for breast cancer overall [odds ratio per standard deviation increment (OR): 2.27, 95% CI: 1.88 to 2.74; PP4: 0.99] and all molecular subtypes except HER2 enriched tumours. A majority of the remaining eight proteins were associated with molecular subtypes characterised by oestrogen-receptor (ER) positive tumours, such as *AOC2*, which was associated with a higher risk of ER-positive [OR: 2.02, 95% CI: 1.48 to 2.75; PP4: 0.73], luminal A [OR: 2.07, 95% CI: 1.51 to 2.83; PP4: 0.99], and luminal B-HER2 negative tumours [OR: 2.54, 95% CI: 1.43 to 4.51; PP4: 0.99]. In contrast, *RALB* was associated with an increased risk of ER-negative [OR: 1.40, 95% CI: 1.20 to 1.64; PP4: 0.93] and HER2 enriched tumours [OR: 1.59, 95% CI: 1.14 to 2.25; PP4: 0.98], as well as breast cancer overall

[OR: 1.16, 95% CI: 1.09 to 1.23; PP4: 0.70]. *GAS1* appeared to have a relatively specific association with risk of triple negative breast cancer [OR: 1.88, 95% CI: 1.42 to 2.47; PP4: 0.86] but was also associated with a lower risk of HER2 enriched breast cancer [OR: 0.46, 95% CI: 0.27 to 0.78; PP4: 0.78]. *MST1* [OR: 1.07, 95% CI: 1.04 to 1.09; PP4: 0.94] was only associated with ER-negative tumours. Four of the 21 proteins (*PDCD6, TLR1, POGLUT3, and LAYN*) identified due to their association with breast cancer were also observed to have colocalised and at least nominal associations with risk for other cancer sites included in this study. Among these, two also associated at nominal significance with ovarian cancer (*TLR1* and *PDCD6*), one also associated with lung cancer (*POGLUT3*) and one associated with each kidney (*POGLUT3*), and bladder cancer (*LAYN*). **Ovary.** One protein had an association with risk of ovarian cancer. *NSF* was associated with multiple ovarian cancer endpoints, including risk of high-grade serous tumours [OR: 2.26, 95% CI: 1.68 to 3.03; PP4: 0.99].

### Upper gastrointestinal and respiratory cancers

**Lung.** We found six proteins (*PTGR1, C4A, MICA/B, SFTPB, NUCB1, POGLUT3*) with strong evidence for an association with lung cancer risk of which two, *SFTPB* and *PTGR1*, were not observed to associate with risk of other cancers. *SFTPB* was associated with lower risk of lung cancer overall [OR: 0.79, 95% CI: 0.69 to 0.91; PP4: 0.82], but also lung adenocarcinoma, and lung cancer in never-smokers. In contrast, *PTGR1* was associated with a lower risk of ever smoking and with squamous cell tumours [OR: 0.79, 95% CI: 0.72 to 0.87; PP4: 0.99]. *C4A* was inversely associated with both risk of lung cancer overall [OR: 0.68, 95% CI: 0.62 to 0.75; PP4: 0.96] and oral cavity cancer [OR: 0.42, 95% CI: 0.29 to 0.59; PP4: 0.99]. **Head and Neck.** *C4A*, *DDX39B*, and *MICA/B* had associations with risk of head and neck cancers of which one, *DDX39B*,

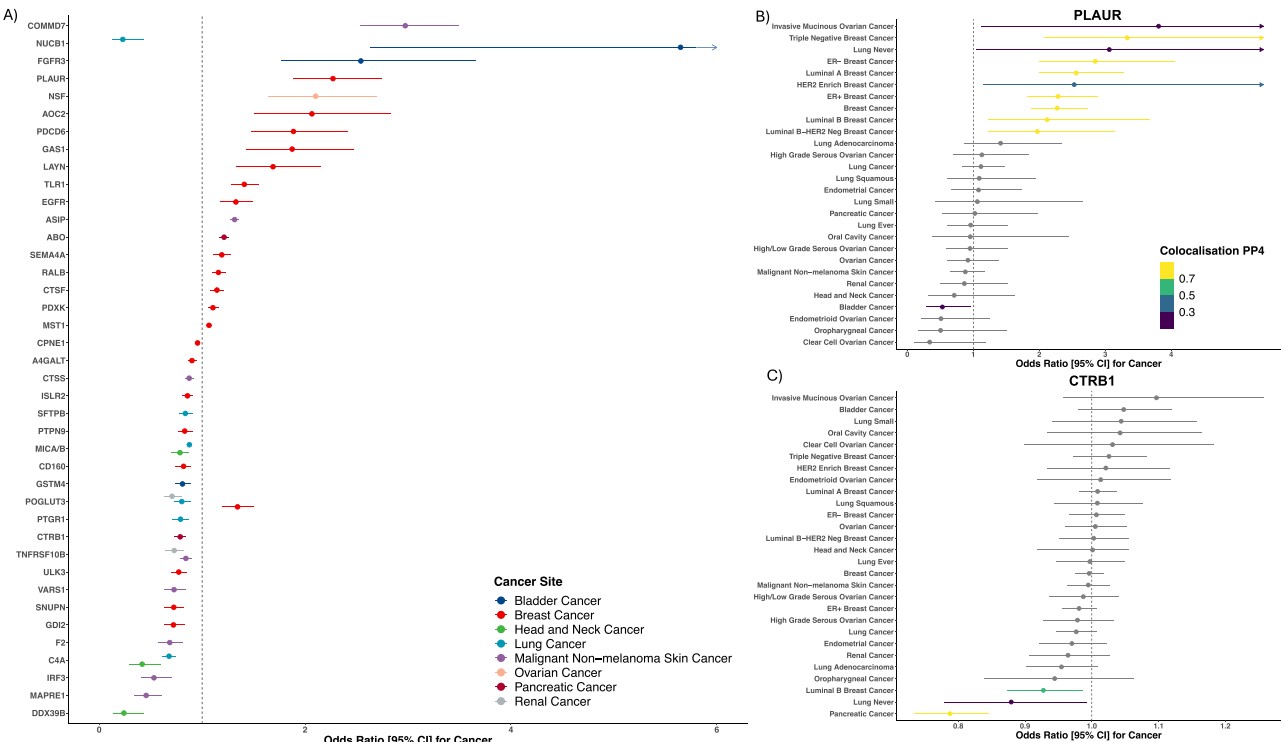

**Fig. 2 | Forrest plot for the associations of 40 protein and cancer risk and pan-cancer risk forrest plots for PLAUR and CTRB1.** Proteins associated with cancer risk (**A**) after correction for multiple testing for the number of proteins analysed in this study per-cancer. Odds ratio estimates are estimated using Wald ratios scaled per standard deviation increment in relative circulating protein concentrations. Confidence intervals (95% CI) are derived using the standard Wald ratio formula and reflect the precision of the cis-pQTL estimate in cancer GWAS scaled by the

beta for the cis-pQTL association with protein concentrations. Sample sizes for cancer GWAS can be found in methods. Association of higher PLAUR with cancer risk (**B**) coloured by the colocalization probability (PP4), where MR Wald *p* < 0.05. Association for higher CTRB1 with cancer risk presented to demonstrate pancreas-specific association (**C**) with colour scheme as described above. All tests are two-sided.

appeared cancer-subtype-specific and was associated with a lower risk of oropharyngeal cancer [OR: 0.24, 95% CI: 0.13 to 0.43; PP4: 0.94]. We also identified an inverse association of *MICA/B* with a lower risk of oral cavity cancer [OR: 0.79, 95% CI: 0.70 to 0.88; PP4: 0.99] and lung cancer overall [OR: 0.88, 95% CI: 0.85 to 0.90; PP4: 0.96]. Conversely, this protein was associated with a modest, nominally significant, increased risk of endometrial cancer [OR: 1.10, 95% CI: 1.05 to 1.15; PP4: 0.89].

### Urological cancers
**Kidney.** *TNFRSF10B* and *POGLUT3* associated with kidney cancer and were also associated with other malignancies. *TNFRSF10B* was associated with a lower risk of both kidney cancer [OR: 0.73, 95% CI: 0.64 to 0.83; PP4: 0.99] and non-malignant melanoma. *POGLUT3* was associated with a lower risk of kidney cancer [OR: 0.71, 95% CI: 0.63 to 0.79; PP4: 0.98] and lung adenocarcinoma [OR: 0.80, 95% CI: 0.73 to 0.89; PP4: 0.99] but an increased risk of both ER- and triple-negative breast cancer. **Bladder.** *NUCB1*, *GSTM4*, and *FGFR3* had associations with bladder cancer risk. One of these, *GSTM4*, was associated with a cancer-specific and lower risk [OR: 0.81, 95% CI: 0.74 to 0.89; PP4: 0.99]. Two others, *NUCB1* [OR: 5.65, 95% CI: 2.64 to 12.09; PP4: 0.98] and FGFR3 [OR: 2.54, 95% CI: 1.77 to 3.66; PP4: 0.99] were both associated with a higher risk of bladder cancer and were observed to have associations with other cancers, including luminal B breast cancer and lung adenocarcinoma.

### Skin and pancreas cancers
**Non-malignant melanoma.** Eight proteins (*TNFRSF10B, F2, CTSS, VARS1, ASIP, IRF3, MAPRE1, COMMD7*) had an association with risk of non-malignant melanoma of which four (*F2, VARS1, IRF3*, and *MAPRE1*) were not observed to associate with other cancers in this study, such as *IRF3* [OR: 0.53, 95% CI: 0.41 to 0.70; PP4: 0.87]. Of the four other proteins, *CTSS* [OR: 0.87, 95% CI: 0.83 to 0.92; PP4: 0.98] and *COMMD7* [OR: 2.97, 95% CI: 2.53 to 3.49; PP4: 0.99] were also associated with lung cancer overall [OR: 0.93, 95% CI: 0.89 to 0.98; PP4: 0.99] and never smoking lung cancer [OR: 2.78, 95% CI: 1.36 to 5.69; PP4: 0.78], respectively. **Pancreas.** *CTRB1* and *ABO* were associated with risk of pancreatic cancer of which one was only associated with pancreas cancer and largely only expressed in the exocrine pancreas, *CTRB1* [OR: 0.79, 95% CI: 0.73 to 0.85; PP4: 0.99]. *ABO* was associated with an increased risk of pancreas [OR: 1.21, 95% CI: 1.17 to 1.26; PP4: 0.97] and endometrial cancers [OR: 1.05, 95% CI: 1.03 to 1.08; PP4: 0.97] but with a lower risk of lung cancer [OR: 0.98, 95% CI: 0.96 to 0.99; PP4: 0.74].

Thirty-eight of the 40 cancer risk proteins were identified only using cis-pQTL for proteins measured on the Somalogic platform. Two proteins (*TNFRSF10B* and *PLAUR*) were identified only using cis-pQTL from GWAS of proteins measured using the Olink platform.

### Replication of candidate aetiological proteins for cancer risk
Replication analyses were conducted for associations where an external cancer GWAS was available. Replication of cis-pQTL MR associations was observed for 29 of the 68 protein-cancer associations that we were able to investigate (Supplementary Data Table 3). For example, primary analyses identified higher *GSTM4* was associated with a lower risk of bladder cancer [OR: 0.81, 95% CI: 0.74 to 0.89, PP4: 0.99] in main analyses, which was replicated in a combined bladder cancer GWAS from UK Biobank and Finngen cohorts [OR: 0.85, 95% CI: 0.77 to 0.93]. Additionally, replication in UKBB and Finngen was observed for other results, including *PLAUR* [breast cancer OR: 1.82, 95% CI: 1.37 to 2.42], POGLUT3 [kidney cancer OR: 0.72, 95% CI: 0.60 to 0.88], and *CTRB1* [pancreas cancer OR: 0.83, 95% CI: 0.75 to 0.92]. No replication GWAS were available for oropharyngeal and high-grade ovarian serous cancer, or luminal A or triple-negative breast cancer.

### Protein association with other traits
**Colocalization and MR PHEWAS.** We identified associations for many proteins, found to associate with cancer risk, with non-cancer endpoints that may be informative for determining the specificity of their associations with cancer or the utility of any potential therapeutic intervention. Notably, however, we did not observe any associations with other traits for *SFTPB*, *EGFR*, and *GAS1*, which were associated with a lower risk of lung adenocarcinoma and a higher risk of breast cancer overall and triple-negative breast cancer, respectively. All results are presented in Supplementary data 4, while two proteins associated with an increased cancer risk are discussed here in greater detail below as emblematic vignettes for the potential consequences of intervening to lower a protein to reduce cancer risk (Fig. 3).

Higher *FGFR3* was associated with an increased risk of bladder cancer. However, we observe that lowering *FGFR3* may potentially have harmful effects on other common sources of morbidity, such as a higher risk of osteoarthritis of the hip or knee [OR: 1.42, 95% CI: 1.24 to 1.63; PP4: 0.99] and a reduced usual walking pace [Beta: −0.05, 95% CI: −0.03 to −0.07; PP4: 0.96], and higher circulating oestradiol levels (SD) [Beta: 0.03, 95% CI: 0.01 to 0.05; PP4: 0.94] and lower circulating albumin (SD) [Beta: −0.09, 95% CI: −0.06 to −0.14; PP4: 0.98]. Nonetheless, lower *FGFR3* may also lead to other potentially beneficial consequences including a higher kidney volume (litres) [Beta: 0.26, 95% CI: 0.14 to 0.38; PP4: 0.96].

Higher *PDXK* was associated with an increased risk of breast cancer. However, lowering *PDXK* was also associated with higher systolic [Beta: 0.72, 95% CI: 0.49 to 0.95; PP4: 0.92] and diastolic blood pressure (mmHg) [Beta: 0.51, 95% CI: 0.38 to 0.64; PP4: 0.96], a higher risk of having hypertension [OR: 1.12, 95% CI: 1.07 to 1.15; PP4: 0.90]. It was also associated with higher eosinophil counts (SD) [Beta: 0.05, 95% CI: 0.03 to 0.08; PP4: 0.98] and diseases of the eye and adnexa [OR: 1.16, 95% CI: 1.08 to 1.23; PP4: 0.79], but a lower risk of reporting hay fever or allergic rhinitis [OR: 0.85, 95% CI: 0.79 to 0.91; PP4: 0.85].

**HyprColoc analyses.** Eight proteins (*A4GALT, ASIP, CTSF, MARE1, PDXK, SEM4A, PLAUR, VARS1*) were observed to have evidence of multi-trait colocalization between the index protein, intermediate phenotypes, and the index cancer endpoint (Supplementary data 5), which may serve to elucidate aetiological pathways to cancer risk. For example, higher *PLAUR*, a breast cancer risk protein, had evidence of a colocalized association with lower monocyte cell count [Beta: −0.52; PP4: 0.99] and higher granulocyte percentage of myeloid white cells [Beta: 0.63; PP4: 0.99], and strong evidence for a shared association of *PLAUR* and these blood cell traits with risk of breast cancer overall [PP4: 0.91], triple negative [PP4: 0.89], luminal A [PP4: 0.90], and ER-negative breast cancer [PP4: 0.89].

**Public Databases.** Thirty-one of our cancer risk proteins had pLI scores of <0.1 implying high tolerance for loss of function variation (Supplementary Data Table 3). Conversely, two proteins, *DDX39B* and *MAPRE1*, appeared highly intolerant of LoF variation and had pLI > 0.9. We observed only limited evidence for the association of pLOF variants in cognate genes for cancer risk proteins with other traits in the UK Biobank using Genebass or the AstraZeneca PheWAS Portal, none of which were cancer endpoints (*LAYN, PTGR1, FGFR3, VARS1, PTPN9*), none of which were cancer endpoints. In Genebass, pLOF in *LAYN* was associated with lower forced expiratory volume in 1-second (beta: −2.99$^{-4}$, $p = 1.91^{-11}$), while variation in *PLAUR* was associated with lower carotid intima-medial thickness (beta: −1.09$^{-1}$, $p = 2.03^{-6}$). *PTGR1* pLOF burden was associated with high monocyte percentage (beta: 5.58$^{-3}$, $p = 1.34^{-7}$), *PTPN9* pLOF burden was associated with recent changes in the speed/amount of moving or speaking (OR: 1.02, $p = 6.84^{-15}$), and *VARS1* with lower mean corpuscular haemoglobin (beta: −1.09$^{-2}$, $p = 2.41^{-6}$). Additionally, rare protein-damaging missense variation in *FGFR3* was the top predictor of osteochondrodysplasia (OR: 61.51,

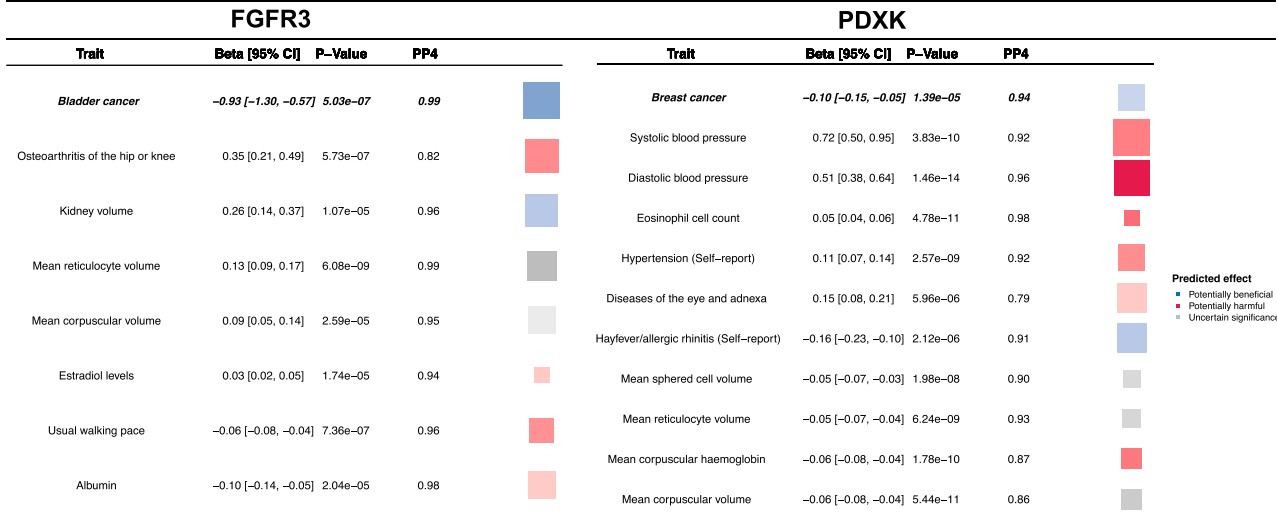

**Fig. 3 | Potential consequences of genetically-predicted protein altering interventions as explemplified using FGFR3 and PDXK.** Potential consequences, presented as beta coefficients and confidence intervals (95% CI) of a protein-lowering intervention for two emblematic cancer risk proteins, FGFR3 and PDXK, that associated with higher risk of bladder and breast cancer, respectively, estimated using Wald ratios for traits where cis-pQTL had a *p*-value for association passing correction for multiple testing in main analyses. For reference, the predicted effect of lowering each protein by 1 SD is present for cancer risk at the top (italics and bold). Below these estimates we present the predicted effect of protein-lowering where colocalization and MR analyses suggest an aetiological link with other traits. Colocalization probability (PP4) is also presented for each trait. All tests are two-sided. Box colour indicates whether a predicted consequence may be beneficial (blue), harmful (red) or have uncertain consequence (grey) on health. Box size is proportionate to absolute beta from MR analyses while box opacity is proportionate to precision of this MR estimate.

7.13[−12]) from analyses in the UK Biobank reported in the AstraZeneca PheWAS Portal.

We also identified evidence of Mendelian disorders associated with genetic alterations to the cognate genes of 17 of the 40 noteworthy proteins, such as *FGFR3* and *F2* in OMIM. Among other conditions, such as stroke and pregnancy loss, *F2* mutations have been associated with higher circulating prothrombin levels and an increased risk of venous thrombosis. Mutations in *FGFR3* have been associated with achondroplasia.

## Drug target pQTL analyses

Harmonised cis-pQTL were available for up to 473 of the total 488 proteins whose cognate gene was mapped to a known drug target for at least one cancer outcome. After correction for multiple testing, we found 18 proteins mapped to drugs that were associated with the risk of at least one cancer endpoint. These included breast [8 proteins], lung [3 proteins], head and neck [2 proteins], kidney [2 proteins], malignant non-melanoma [3 protein], and bladder [1 protein], and pancreas [1 protein] cancers that were the target of at least one pharmaceutical intervention (Supplementary data 6). We additionally identified eight proteins that were the target of a drug under investigation at phase I clinical trials or higher, while eight out of these proteins are currently at the preclinical stage or biological testing, which may imply their drug-ability is under active evaluation. Fifteen of the proteins we identify associated with cancer risk, including *SFTPB* and *GAS1*, did not appear to be under active current investigation as a drug target (Fig. 4.).

Notably, a majority of drugs targeting cancer risk proteins were typically either small molecular inhibitors (SMIs) or monoclonal antibodies, some of which are used for the treatment of the cancer indicated by the risk association. For example, higher *FGFR3* was associated with an increased risk of bladder cancer. *FGFR3* is directly inhibited by Erdafitinib[19], which is a SMI and a treatment of urothelial cancers. We also identified an association of *EGFR* with a higher risk of breast cancer, which is inhibited by a variety of monoclonal antibodies in the treatment of breast cancer[20].

## Discussion

Using genetic data from up to 337,822 cancer cases, we have identified 40 proteins with a likely role in the aetiology of at least one type of cancer. Most proteins that we identify associated with cancer risk were replicated using an independent cancer GWAS and had not been reported on before in this context. Some risk proteins were associated with a potential causal role in specific molecular subtypes of cancer, some were risk proteins with specific expression in the relevant organ, while others associated with the risk of multiple cancers. We also identified proteins with a potential aetiological role in cancer risk that also mediate the therapeutic effects of specific drugs that may lead to possible drug repurposing. Furthermore, we identified proteins that are associated with cancer but do not appear to be under active investigation, indicating that they may represent opportunities to develop new therapeutic treatments of cancer.

Proteins are crucial for the maintenance of cellular structure and regulation of cell signalling involved in all human biological processes. Identifying protein markers of cancer aetiology may inform our understanding of the pathways to tumorigenesis and serve as a fruitful tool for discovering biomarkers of cancer risk. As a proof-of-concept for how genetic methods may identify causal cancer genes we used this approach to highlight a causal role of *EGFR* in breast cancer risk. *EGFR* is a COSMIC[21] consensus gene associated with breast cancer risk. Among other processes, *EGFR* has an established role in cell proliferation, migration and differentiation that facilitates the uncontrolled division of cancer epithelial cells[22], including those in the mammary glands[20]. *EGFR* may therefore serve as a potential positive control for protein MR as a method to discover oncogenic pathways.

We also identified a role for proteins with tissue-specific expression at the site of the cancer indicated in risk analyses. One example is *CTRB1*, which is a serine protease digestive enzyme precursor produced largely by the exocrine pancreas and which was associated with a lower risk of pancreatic cancer in two independent cancer GWAS. Along with *PRSS2* and *SPINK*, *CTRB1* acts to degrade trypsinogen in the pancreas. Mutations in *CTRB1* have been associated with higher

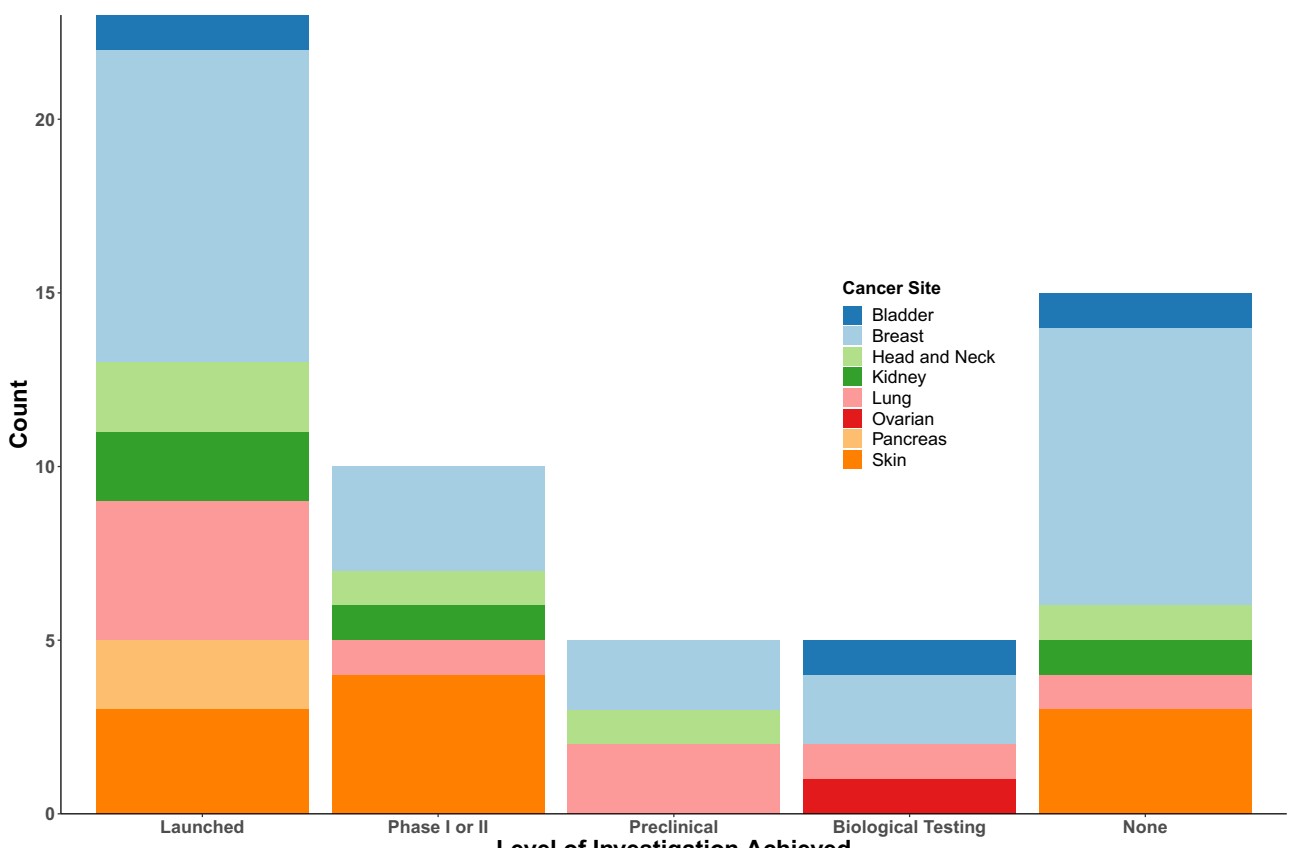

**Fig. 4 | Stacked bar plot that depicts identified proteins associated with cancer by their current highest level of therapeutic investigation, which additionally has colours that stratified results by cancer site.** Figure displays a stacked bar chart, coloured by cancer endpoint for protein in question, mapping of cancer-risk proteins to currently available information on clinical/pharmaceutical investigations. These include cancer-risk proteins that are the target of a currently launched drug, under investigation at Phase I or II, under preclinical or biological testing, and those not known to be under current investigation.

intrapancreatic trypsin activation, chronic inflammation, and loss of pancreas function[23]. Further, *CTRB1* expression has been found in the topmost downregulated genes in matched tumour-normal pancreas samples[24]. However, the *CTRB1* region on chromosome 16 contains various complex genomic rearrangements that include a short deletion in *CTRB2* associated with pancreas cancer risk[25] and a 16.6kb inversion between *CTRB1* and *CTRB2*, which is associated with chronic pancreatitis[26], a risk factor for pancreatic cancer. While we do not find an association of *CTRB2* with pancreatic cancer risk, we do add to previous evidence[27] that *CTRB1* may affect the risk of both type 1 and type 2 diabetes in PHEWAS analyses, which may be risk factors for pancreas cancer. Therefore, further analyses are needed to clarify the role of the *CTRB1-CTRB2* locus in pancreatic cancer risk, which could include pre-clinical assessment of *CTRB1* and *CTRB2* as potential anti-tumour agents as was previously conducted for pro-enzymes *PRSS1* and Chymotrypsinogen A[28].

Notably, among cancer-risk proteins we observe a high tolerance for haploinsufficiency and a general absence of congruent evidence using whole exome analyses of potential loss of function variants in the UK Biobank for the role of proteins we identify with a risk of cancer. We suspect this may at least in part be due to study power for whole exome analyses and challenges in the interpretation of pLI scores in the context of cancer[29]. However, it may also suggest these proteins affect cancer risk via regulation of abundance as compared to the presence or absence of functional gene copies. Moreover, this finding may imply protein-truncating alterations to these proteins' cognate genes are unlikely to lead to severe disease in early life, which may affect their relevance in therapeutic prevention.

Proteins are essential targets for drug development[30]. However, not all proteins identified as risk factors for cancer are suitable therapeutic targets. Biochemical factors that contribute to a protein's druggability include protein-class (membrane-bound[19,31], or soluble and secreted ligands[32], for example), current understanding of its pathophysiology and molecular function, its cellular location, and the presence of drug binding sites. Epidemiological factors that contribute to a protein's utility as a therapeutic target include the magnitude and specificity of its association with cancer risk, the expected incidence, morbidity and mortality of the cancer of interest, and our current ability to robustly identify an at-risk population to receive a proposed therapeutic intervention. We present three vignettes to illustrate both the potential and the complexities when investigating proteins for cancer prevention.

### FGFR3 and bladder cancer
We found an increase in *FGFR3* associated with a more than two-fold increased risk of bladder cancer in two independent cancer GWAS. *FGFR3* is an established oncogene with a well-described role in proliferative signalling and evasion of cell death[22]. Further, *FGFR3* is the target of an approved tyrosine kinase inhibitor used to treat urothelial cancer, erdafitinib[33], which has modest reported toxicity[34]. Whereas these data may suggest *FGFR3* as a potential target for therapeutic prevention, further complementary analyses suggested lower *FGFR3* may increase the risk of osteoarthritis, a common source of morbidity among older populations, and rare variant studies show that damaged *FGFR3* increases the risk of bone disorders. Considering bladder cancer is not among the most common cancers and has a relatively good

prognosis, it may therefore not be justified to target *FGFR3* specifically to prevent bladder cancer. This example highlights the importance of considering a broad spectrum of potential health outcomes when developing therapeutic prevention strategies for specific cancer sites to identify potential unintentional but harmful secondary effects.

### SFTPB and lung cancer

We found that higher *SFTPB* was associated with a lower risk of lung adenocarcinoma. *SFTPB* is produced specifically by alveolar type II cells[35] and is essential for healthy lung function[36,37]. Previous studies identified downregulation of *SFTPB* in mice to be associated with accelerated tumour growth and rate of epithelial-mesenchymal transition[38], and higher *SFTPB* expression in tumour samples is associated with better survival among lung cancer patients[37]. Notably, we did not find evidence of non-lung cancer associations for *SFTPB*, though an association with chronic obstructive pulmonary disease has previously been reported[39]. Lung cancer is a highly incident cancer and the leading cause of cancer death globally. Multiple tools for the identification and referral of individuals at high risk of lung cancer for screening programmes are in current use with additional biomarker-based risk models under development, such as within the INTEGRAL programme[40]. Lung cancer may therefore present an appealing target for therapeutic prevention as high-risk individuals who may benefit from such interventions are readily identifiable if a suitable agonist for *SFTPB* can be identified.

### GAS1 and triple-negative breast cancer

*GAS1*, which was associated with an increased risk of triple negative breast cancer (TNBC), is an essential co-receptor of hedgehog (Hh) signalling and specifically expressed at high levels in healthy fibroblasts[41,42]. Notably, we did not observe evidence that *GAS1* had a similar effect on the risk of other molecular subtypes of breast cancer or on non-cancer phenotypes, which may imply *GAS1* has a specific effect to increase TNBC risk. *GAS1* has a critical role, via a structural interaction, in the delivery of sonic hedgehog to its downstream receptor, *PTCH1*[43]. Furthermore, Hh-activated TNBC mouse models indicated a specific ligand-dependent paracrine role for the Hh pathway acting via cancer-associated fibroblasts (CAF) to maintain chemotherapy-resistant breast cancer stem cell phenotypes. An independent study also identified a paracrine Hh pathway signature was associated with a higher risk of metastasis and breast cancer-specific death in triple-negative disease[44]. TNBC is an aggressive molecular subtype of breast cancer characterised by resistance to many established treatments[45]. Therefore, compared to currently available drugs that target the SMO gene in any cell type, *GAS1* antagonism could provide a therapeutic approach for CAF-specific Hh pathway inhibition in chemoresistant TNBC[46].

Our study has limitations that should be acknowledged. Firstly, our study was not comprehensive in evaluating the entire proteome as we are limited to proteins that have been measured using blood-based multiplex affinity platforms. Secondly, where we have cis-pQTL, our power to discover associations may be limited by cancer GWAS sample size and the heritability of the cancer itself. This may partly explain the discrepancy in the number of risk proteins identified for different cancer sites and the ability to replicate protein associations with cancer risk. We also note that colocalization can be sensitive to the presence of complex genetic architecture or loci where multiple causal signals exists but where only a subset is shared between traits. Similarly, it can be difficult to interpret colocalization between a protein and cancer risk where a cognate gene sits in a genomic region with irregular haplotype structure, such as *C4A* associations with lung cancer on chromosome six, or when two genes that lie immediately adjacent to each other, both share similar associations with cancer risk, such as *PTPN9* and *SNUPN* with breast cancer risk in our analyses. We also note that our replication analyses leveraged data from cancer

GWAS in a Finnish population, and thus, that well-documented changes in allele frequencies due to founder effects may have hindered successful replication for some proteins. Finally, we note that our results are restricted to participants of European ancestry due principally to the current availability of protein GWAS; while we are aware of a recent protein GWAS in the ARIC cohort[47] in people of African American descent, we did not have access to large cancer GWAS for endpoints in this study to utilise these data.

Conversely, our study had several notable strengths. Our widespread integration of colocalization with cis-pQTL MR analyses demonstrated that upwards of 70% of proteins with a nominally significant MR association did not have support for a shared causal locus. While this is closer to 50% for proteins passing our Bonferroni threshold, we suggest that these results demonstrate the importance of presenting colocalization in parallel with any cis-pQTL MR analysis. The inclusion of multiple cancer endpoints in our study has also allowed us to identify proteins with both associations across multiple cancer endpoints and those with cancer-specific associations. Similarly, we conducted analyses and integrated multiple sources of clinically relevant data to better contextualise which of the 40 risk proteins are associated with other non-cancerous traits. Additionally, we mapped our results to established drug targets as well as proteins currently undergoing investigation as therapeutics. In doing so, we highlight opportunities for drug repurposing, but note that risk proteins that are not currently targeted by any drug may serve as appealing targets for future drug development.

The increasing availability of multi-omic data generated in large biobanks and cancer consortia presents an unprecedented opportunity to leverage large-scale genetic data to discover disease pathways. We present an expansive assessment of 2074 proteins in relation to nine common cancer sites, and present multiple proteins implicated in the aetiology of specific cancer types. We demonstrate the importance of carrying out complementary analyses to characterise the disease specificity and pleiotropy that is present for many proteins. We believe these results bring important context both when understanding aetiological pathways and potential adverse consequences of any potential protein-altering intervention. After further experimental and functional follow-up, as well as other pre-clinical and clinical investigations, some of the forty proteins we have identified may provide new options for prevention and be part of the clinical strategies needed to limit the expected increase in cancer incidence.

## Methods

This work used summary genetic association data from previously published GWAS. All studies contributing data to these analyses had the relevant institutional review board approval from each country and all participants provided informed consent.

### Overall study design

We sought to identify aetiological proteins for nine common cancers, including cancer of the head and neck, lung, kidney, pancreas, bladder, breast, ovary, and endometrium, as well as malignant non-melanoma. We used MR to evaluate the association of 2074 blood protein concentrations with cancer risk based on cis-pQTL single nucleotide polymorphisms (SNPs). We subsequently performed colocalization analyses for loci where MR indicated a nominally significant association with cancer risk, to assess the presence of confounding by linkage disequilibrium (LD). Where other independent sources of cancer GWAS were available, we also performed an external validation of cis-pQTL associations with cancer risk. We then conducted an MR and colocalization PHEWAS as well as a review of public databases to assess whether proteins identified in our analyses as cancer risk factors were also associated with other important characteristics or diseases, which may inform potential adverse effects of future

**Table 1 | Description of cancer risk GWASs including ICD-10 codes, case and control counts, and study reference**

| Cancer | ICD-10 | Case/Control | ≥80% power to detect:[c] | Ref |
|---|---|---|---|---|
| Head and neck[a] | C02.0–C02.9, C03.0–C03.9, C04.0–C04.9, C05.0–C06.9, C01.9, C02.4, C09.0–C10.9 | 6034/6,585 | 1.55 | 56 |
| Pancreas | C25 | 7638/7364 | 1.50 | 57 |
| Lung[a,b] | C34 | 41,477/105,297[b] | 1.14 | 58 |
| Malignant non-melanoma | C43 | 23,694/372,016 | 1.17 | 59 |
| Breast[a] | C50 | 133,384/113,789 | 1.11 | 60 |
| Endometrium | C54.1 | 12,906/108,979 | 1.23 | 61 |
| Ovary[a] | C56 | 25,509/40,941 | 1.22 | 62 |
| Kidney | C64 | 10,784/20,406 | 1.33 | 63 |
| Bladder | C67 | 8988/11,978 | 1.40 | 64 |

[a]Cancer subtypes included and described in Supplementary Methods.
[b]Total effective sample size from a meta-analysis between GWAS of family history and GWAS in INTEGRAL-ILCCO.
[c]Estimate for the expected detectable risk association based on median variance explained (1.3%) in cis-pQTL included in analyses. Calculated using https://cnsgenomics.shinyapps.io/mRnd/.

protein-altering interventions. Finally, we mapped cancer risk proteins to targets for approved drugs and those being evaluated in ongoing clinical trials.

## Protein effects on cancer risk

**Collection and quality control for cis-pQTL.** We gathered summary statistics from publicly available protein GWAS' and extracted all independent SNPs associated (at least $p < 5 \times 10^{-08}$) with a protein concentration in blood and lying within 1 megabase of a protein's cognate gene (referred to as cis-pQTL), with clumping at $r^2 < 0.01$ and with a minor allele frequency ≥0.01. We additionally re-processed previously published protein GWAS on the OpenGWAS platform to identify additional cis-pQTL significant at $p < 5 \times 10^{-05}$ due to strong a priori for SNP associations with protein concentrations at or near by its cognate gene (See Supplementary data 1 and Supplementary Methods). Only cis-pQTL with an $F$ statistic > 10 were taken forward for analysis. *Cancer GWAS summary statistics.* Nine cancer outcomes and their subtypes (where applicable/available) were considered in this study, including cancer of the bladder, breast, endometrium, head and neck, lung, ovary, pancreas, kidney, and malignant non-melanoma (Table 1. and Supplementary Methods). Estimated power calculations are presented in Table 1 based on median variance explained in protein levels by cis-pQTL (1.3%).

**MR and colocalization analyses.** Cis-pQTL were harmonised with GWAS summary statistics for each cancer outcome by matching on rsID where directly available, and by selecting a proxy where necessary. Primary risk estimates were odds ratios, as well as their accompanying $p$-values, estimated using per-cis-pQTL Wald Ratios. All MR associations with $p_{Wald} < 0.05$ were subsequently evaluated for confounding by LD and the probability of a shared causal locus (PP4) between protein concentrations and cancer risk using two approaches: conventional colocalization[18,48] and conditional iterative colocalization[18]. The greatest PP4 from these two colocalization methods [PP4$_{max}$] was used and we considered PP4$_{max}$ > 0.7 as indicating cis-pQTL MR associations were unlikely to have been confounded by LD. Cis-pQTL with Bonferroni significant associations (i.e., $p_{Wald} < 0.05/N_{Proteins}$ where $N_{Proteins}$ is the number of unique proteins analysed for a given cancer outcome) that also had evidence of colocalization (i.e., PP4$_{max}$ > 0.7) were subjected to further follow-up analyses. Further details in Supplementary Methods.

**Replication of candidate aetiological proteins for cancer risk.** Where data were available, we conducted a replication of noteworthy cis-pQTL MR associations (i.e. PP4$_{max}$ > 0.7 & Bonferroni $p_{Wald}$) using external cancer GWAS data from either a meta-analysis of FinnGen r9[49] and the UK Biobank[50], or from FinnGen alone depending on the

endpoint (see Supplementary Data Table 3 for case counts and Supplementary Methods).

**Reverse MR analyses.** We assessed the potential impact of cancer risk on protein levels for proteins identified in main analyses using the inverse-variance weighted and weighted median approaches and using GWAS significant [$p < 5 \times 10^{-08}$] and independent genetic variants [clumping at $r^2 < 0.01$]. Additionally, we used the Egger intercept to assess directional pleiotropy and, were it indicated significant pleiotropy, we also conducted MR-Egger analyses. Data were harmonised as described in protein-cancer analyses. We required consistent evidence from all risk estimation methods and Bonferroni significance corrected for the number of proteins investigated to support the association of cancer risk with protein levels.

## Protein associations with other traits

We conducted additional analyses to provide greater context to the specificity of an identified cancer risk protein association using PHE-WAS MR and colocalization analyses as well as consulting several public databases. We performed these steps to collate information on potential harmful or additional beneficial consequences of altering identified protein concentrations in human populations. Firstly, we assessed the association of each cancer risk protein with all available traits on the OpenGWAS platform using MR and colocalization methods as previously described[51]. We additionally used HyprColoc[52] to assess whether any subset of protein-trait associations we identified may elucidate a potential causal pathway between the risk of cancer and the indicated risk protein. Secondly, we conducted a search of several relevant databases that included information on probability of loss of function intolerance (pLI), exome-sequencing studies, rare-variant association studies, and Mendelian genetics not likely to overlap with OpenGWAS traits. Further details in Supplementary Methods.

## Drug target pQTL analyses

A secondary analysis was conducted restricting MR and colocalization analyses to cis-pQTL with a cognate gene that is an established pharmaceutical target for the action of one of 867 existing drugs identified by reference to a combination of databases (including Drugbank and ClinicalTrials.gov) and expert curation (see Supplementary Methods). We defined noteworthy drug target proteins as having $p_{Wald} < 0.05/N_{Proteins}$ where $N_{Proteins}$ is the number of unique proteins analysed for a given cancer outcome that were identified as the cognate gene of a pharmaceutical target. We additionally queried the Cortellis database (https://www.cortellis.com/ Clarivate Analytics) to assess the highest current level of clinical development stage for proteins identified to associate with cancer risk from main analyses.

Analysis was performed using R version 4.2.1, tidyverse(2.0.0), ggplot2(3.4.4), TwoSampleMR package (0.5.6), Coloc package (5.2.3), plink (1.9).

## Reporting summary

Further information on research design is available in the Nature Portfolio Reporting Summary linked to this article.

## Data availability

Summary statistics from Zheng et al.[53] can be obtained from Open-GWAS (https://gwas.mrcieu.ac.uk/), from Folkersen et al.[54] at http://www.scallop-consortium.com, from Ferkingstad et al.[55] at https://www.decode.com/summarydata/, and from Pietzner et al.[15] at https://omicscience.org. We obtained summary genetic association data on breast cancer risk from the Breast Cancer Association Consortium (https://bcac.ccge.medschl.cam.ac.uk/), ovarian cancer risk from the Ovarian Cancer Association Consortium (https://ocac.ccge.medschl.cam.ac.uk/), and endometrial cancer risk from the Endometrial Cancer Association Consortium (https://www.ebi.ac.uk/gwas/publications/30093612#study_panel). Approval was received to use restricted summary genetic association data from INTEGRAL ILCCO consortia after submitting a proposal to access this data. Summary genetic association data from these consortia can be accessed by contacting INTEGRAL ILCCO (rayjean.hung@lunenfeld.ca) (https://ilcco.iarc.fr). Approval was also received to use restricted summary genetic association data on pancreatic cancer risk via dbGaP release phs000206.v5.p3. To enquire about gaining access to summary genetic association data for renal and head and neck cancer risk, contact brennanp@iarc.fr. To enquire about gaining access to summary genetic association data for bladder cancer risk, contact bart.kiemeney@radboudumc.nl. Summary statistics for Malignant non-melanoma were obtained from OpenGWAS (https://gwas.mrcieu.ac.uk/). Source data are provided with this paper.

## Code availability

Code for the data analyses can be found at https://github.com/karlsmithbyrne/Pan_Cancer_Protein_MR_2024/blob/main/Manuscript_COde.

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

## Acknowledgements

K.S.B., J.A., T.D. and R.T. were supported by the Cancer Research United Kingdom (grant no. C8221/A29017). H.B.S. and A.V.S. are funded by The Swedish Cancer Society (Grants 20 0990). R.M.M. is a National Institute for Health Research Senior Investigator (NIHR202411). R.M.M. is supported by a Cancer Research UK 25 (C18281/A29019) programme grant (the Integrative Cancer Epidemiology Programme). RMM is also supported by the NIHR Bristol Biomedical Research Centre which is funded by the NIHR (BRC-1215–20011) and is a partnership between University Hospitals Bristol and Weston NHS Foundation Trust and the University of Bristol. RMM is affiliated with the Medical Research Council Integrative Epidemiology Unit at the University of Bristol which is supported by the Medical Research Council (MC_UU_00011/1, MC_UU_00011/3, MC_UU_00011/6, and MC_UU_00011/4) and the University of Bristol. Department of Health and Social Care disclaimer: The views expressed are those of the author(s) and not necessarily those of the NHS, the NIHR or the Department of Health and Social Care. **Disclaimer:** Where authors are identified as personnel of the International Agency for Research on Cancer/World Health Organization, the authors alone are responsible for the views expressed in this article and they do not necessarily represent the decisions, policy, or views of the International Agency for Research on Cancer/World Health Organization.

## Author contributions

K.S-B., A.M., R.C.T., M.J., Å.H. conceived of and designed the study. K.S-B. performed data analyses. K.S-B., Å.H., M.D., T.D., A.V.S., H.B.S., M.K, M.P., C.L., J.A., R.C.P., J.M., P.B., S.Z., B.R., J.Y., R.M.M., J.B., X.J.M., A.B., X.S., J.W., T.L.A., R.J.H., C.A., M.P., N.R., S.C., R.C.T., M.J., A.M. drafted the manuscript. All authors read and approved the final manuscript.

## Competing interests

Å.H., M.D., J.B., X.J.M. and A.M. are employees at Pfizer Inc., the remaining authors declare no competing interests.

## Additional information

[1]Cancer Epidemiology Unit, Oxford Population Health, University of Oxford, Oxford, UK. [2]External Science and Innovation, Pfizer Worldwide Research, Development and Medical, Stockholm, Sweden. [3]Department of Medicine, Karolinska Institute, Stockholm, Sweden. [4]Department of Surgical Sciences, Functional Pharmacology and Neuroscience, Uppsala University, Uppsala, Sweden. [5]MRC Epidemiology Unit, University of Cambridge, Cambridge, UK. [6]Computational Medicine, Berlin Institute of Health at Charité – Universitätsmedizin Berlin, Berlin, Germany. [7]Precision Healthcare Institute, Queen Mary University of London, London, UK. [8]Genomic Epidemiology Branch, International Agency for Research on Cancer (IARC-WHO), Lyon, France. [9]Department of Human Genetics, McGill University, Montréal, QC, Canada. [10]Departments of Medicine (Endocrinology), Human Genetics, Epidemiology and Biostatistics, McGill University, Montréal, QC, Canada. [11]MRC Integrative Epidemiology Unit, University of Bristol, Bristol, UK. [12]Population Health Sciences, Bristol Medical School, University of Bristol, Bristol, UK. [13]NIHR Bristol Biomedical Research Centre, Hospitals Bristol and Weston NHS Foundation Trust and the University of Bristol, Bristol, UK. [14]Cancer Immunology Discovery, Pfizer Worldwide Research and Development Medicine, Pfizer Inc, San Diego, USA. [15]Oncology Research Unit, Pfizer Worldwide Research and Development Medicine, Pfizer Inc, San Diego, USA. [16]Department of Public Health and Primary Care, University of Cambridge, Cambridge, UK. [17]Usher Institute, MRC Human Genetics Unit, University of Edinburgh, Edinburgh, UK. [18]Division of Cardiovascular Medicine and the Cardiovascular Institute, School of Medicine, Stanford University, Stanford, USA. [19]Prosserman Centre for Health Research, Lunenfeld-Tanenbaum Research Institute, Sinai Health System and University of Toronto, Toronto, Canada. [20]Department of Medicine, Epidemiology Section, Institute for Clinical and Translational Research, Baylor Medical College, Houston, USA. [21]Division of Cancer Epidemiology and Genetics, National Cancer Institute, Rockville, USA. [22]Occupational and Environmental Epidemiology Branch, Division of Cancer Epidemiology and Genetics, National Cancer Institute, Rockville, USA. [23]These authors contributed equally: Ruth C. Travis, Mattias Johansson, Anders Mälarstig. ✉e-mail: karl.smith-byrne@ndph.ox.ac.uk

