## [Peer Review File · Nature Communications]

REVIEWER COMMENTS

Reviewer #1 (Remarks to the Author): expertise in pan-cancer GWAS

The manuscript entitled “Identifying therapeutic targets for cancer: 2,094 circulating proteins and risk of nine cancers” utilized nine common cancers GWAS and protein QTL public datasets to address the role of circulating proteins in cancer risk. Unfortunately, the manuscript is too descriptive without in-depth analyses. Moreover, many of the methods, analyses and results were not clear.

1. The datasets did not be clearly described and quality control is hard to evaluate. For example, the authors should have some quality control analysis of GWAS summary statistics. In addition, why the authors use two different cutoffs ($p < 5e-8$ for protein GWAS, $p < 5e-5$ for openGWAS)?
2. It is unclear how the p-value is calculated in Table 2.
3. The result is not robust by only using MR method to identify causal effects of proteins associated with cancer risk. The authors should validate their finding using other approaches.
4. The authors should discuss the clinical implications of 40 proteins associated with cancer risk such as sensitivity analysis.
5. Horizontal pleiotropy and heterogeneity are major issues for Mendelian randomisation (MR) studies, it remains unclear how the authors resolved heterogeneity and corrected for potential directional horizontal pleiotropy between exposure and outcome.

Reviewer #2 (Remarks to the Author): expertise in Mendelian randomisation and colocalization analysis

The authors presented an interesting and thorough analysis of a very large sample size, where they investigated the causal circulating proteins of nine common cancer risks based on proteome-wide MR and colocalization. Additional phenome-wide analyses were then performed to assess the associations of proteins with other traits and to identify the potential drug targets. In general, this manuscript is a well-executed study and the results have merit.

However, I have a few comments regarding the paper.

1. Does it make sense to use $5e-5$ instead of $5e-8$ as the significance threshold in the criteria for selection of IVs? It is necessary to describe the threshold selection.
2. The author demonstrated that some risk proteins were associated with the risk of multiple cancers, or may have a role in potential adverse outcomes. I am concerned about whether there are pleiotropy instrumental variables (cis-pQTL) in proteome-wide MR and should be considered.
3. If there are more than 3 instruments, it would be nice to see the Egger-based, median-based MRs as sensitivity analysis, and MR-PRESSO for more than 4 IVs.
4. Reverse MR between cancers and risk proteins could be added for more analysis.

5. F statistic could be calculated to measure the strength of each instrument.
6. Please estimate the statistical power of the MR association, which is implemented in mRnd (<https://cnsgenomics.shinyapps.io/mRnd/>).
7. It could be more convincing to validate the findings using pQTLs data from an independent cohort.
8. Based on the central dogma, is there a consistency between plasma protein and blood gene?
9. More details on the clinical value of risk proteins could be added to the discussion.

REVIEWER COMMENTS

Reviewer #1

The manuscript entitled “Identifying therapeutic targets for cancer: 2,094 circulating proteins and risk of nine cancers” utilized nine common cancers GWAS and protein QTL public datasets to address the role of circulating proteins in cancer risk. Unfortunately, the manuscript is too descriptive without in-depth analyses. Moreover, many of the methods, analyses and results were not clear.

1. *The datasets did not be clearly described and quality control is hard to evaluate. For example, the authors should have some quality control analysis of GWAS summary statistics.*

Author Response: We thank the reviewer for their comments. We have now included further details on the protein GWAS used in our study in the supplementary materials. Additionally, we have filtered our cis pQTL to exclude variants with a minor allele frequency (MAF) < 0.01 as described on line 197:

We gathered summary statistics from publicly available protein GWAS' and extracted all independent SNPs associated (at least $p < 5e-08$) with a protein concentration in blood and lying within 1 megabase of a protein's cognate gene (referred to as cis pQTL), with clumping at $r^2 < 0.01$ and with a minor allele frequency ≥ 0.01 .

After applying this MAF filter, we excluded 20 proteins from our analysis meaning that we now report results for 2,074 proteins in our manuscript. **However, none of the 20 proteins had support from both MR analyses after correction for multiple testing and colocalization, and so none of our main cancer risk proteins were affected.**

We have also added text that describes the filtering for instrument strength by F statistic > 10 to the main methods (previously in the supplementary methods), which is on line 203:

Only cis pQTL with an F statistic > 10 were taken forward for analysis.

2. *In addition, why the authors use two different cutoffs $p < 5e-8$ for protein GWAS, $p < 5e-5$ for openGWAS?*

Author Response: While many GWAS represent an agnostic hypothesis-generating investigation into the genetic determinants of a trait, GWAS of the circulating proteome seek to identify genetic determinants that fall into two categories: *trans* pQTL that lie outside the proteins' cognate gene and *cis* pQTL that lie nearby (typically 1 megabase) or within the proteins' cognate gene. A GWAS for *cis* pQTL is, however, not genome-wide and instead is a highly restricted search for genetic variants that lie within a predetermined region set by the strong biological *a priori* that variants that are located close to a proteins' cognate gene may associate with a proteins' plasma levels. We therefore join previous studies (<https://www.medrxiv.org/content/10.1101/2021.08.03.21261494v1.full.pdf>, PMID: 32895551) in assigning a more lenient threshold to identify *cis* pQTL to reflect the substantially lower burden of tests that are conducted in the process of their identification. In our study, as in a previous SCALLOP publication referenced above, we chose the common $5e-5$ suggestive genome-wide threshold for this purpose and used it to retrieve additional *cis* pQTL from two studies that had full summary statistics that were available to query within the openGWAS platform.

3. *It is unclear how the p-value is calculated in Table 2.*

Author Response: We outline in the methods section on page 5 that “Primary risk estimates were odds ratios estimated using per-cis pQTL Wald ratios”, and the p-values in Table 2 were taken from these cis-pQTL MR analyses using the Wald ratio method. We have updated text for clarity on line 211, which now reads:

Primary risk estimates (odds ratios), as well as their accompanying p-values, were estimated using per-cis pQTL Wald Ratios.

4. *The result is not robust by only using MR method to identify causal effects of proteins associated with cancer risk. The authors should validate their finding using other approaches.*

Author Response: We thank the reviewer for their comment and agree that a single MR analysis would not be sufficient to rule out a number of important biases that may confound the identification of aetiological proteins for cancer risk. For this reason, we additionally conducted colocalization analyses to assess the probability that our findings were confounded by linkage disequilibrium and, where additional independent cancer summary statistics were available, we performed external validation of our initial MR findings. It is our hope that future studies will seek to further validate our findings using potential functional or experimental validation, such as with the pre-clinical study of CTRB1 and its role in pancreas cancer, but any such analyses fell outside the scope of our design.

5. *The authors should discuss the clinical implications of 40 proteins associated with cancer risk such as sensitivity analysis.*

Author Response: We thank the reviewers for highlighting the important questions of clinical implications from our work. We have provided three vignettes to describe possible routes to translation for protein-MR findings for cancer prevention. After further experimental and functional follow-up, as well as other pre-clinical and clinical investigations, some of these forty proteins may provide new options for prevention and be part of the strategies needed to limit the expected increase in cancer incidence. We have now added a sentence to the discussion to further comment on these potential clinical implications of our work on line 590:

After further experimental and functional follow-up, as well as other pre-clinical and clinical investigations, some of the forty proteins we have identified may provide new options for prevention and be part of the clinical strategies needed to limit the expected increase in cancer incidence.

6. *Horizontal pleiotropy and heterogeneity are major issues for Mendelian randomisation (MR) studies, it remains unclear how the authors resolved heterogeneity and corrected for potential directional horizontal pleiotropy between exposure and outcome.*

Author Response: We agree that horizontal pleiotropy is an important concern in Mendelian randomisation studies. We chose to exclude trans-pQTL from our study design due to their highly pleiotropic nature and ambiguous link to established regulatory roles in circulating protein concentrations. Cis pQTL are much less vulnerable to the influence of horizontal pleiotropy as they sit in or nearby a protein’s cognate gene, and likely impact protein levels directly via its altered transcription or translation (PMID: 35527238).

Reviewer #2 (Remarks to the Author):

The authors presented an interesting and thorough analysis of a very large sample size, where they investigated the causal circulating proteins of nine common cancer risks based on proteome-wide MR and colocalization. Additional phenome-wide analyses were then performed to assess the associations of proteins with other traits and to identify the potential drug targets. In general, this manuscript is a well-executed study, and the results have merit. However, I have a few comments regarding the paper.

1. Does it make sense to use 5e-5 instead of 5e-8 as the significance threshold in the criteria for selection of IVs? It is necessary to describe the threshold selection.

Author Response: While many GWAS represent an agnostic hypothesis-generating investigation into the genetic determinants of a trait, GWAS of the circulating proteome seek to identify genetic determinants that fall into two categories: *trans* pQTL that lie outside the proteins' cognate gene and *cis* pQTL that lie nearby (typically 1 megabase) or within the proteins' cognate gene. A GWAS for *cis* pQTL is, however, not genome-wide and instead is a highly restricted search for genetic variants that lie within a predetermined region set by the strong biological *a priori* that variants that are located close to a proteins' cognate gene may associate with a proteins' plasma levels. We therefore join previous studies (<https://www.medrxiv.org/content/10.1101/2021.08.03.21261494v1.full.pdf>, PMID: 32895551) in assigning a more lenient threshold to identify *cis* pQTL to reflect the substantially lower burden of tests that are conducted in the process of their identification. In our study, as in a previous SCALLOP publication referenced above, we chose the common 5e-5 suggestive genome-wide threshold for this purpose and used it to retrieve additional *cis* pQTL from two studies that had full summary statistics that were available to query within the openGWAS platform.

2. The author demonstrated that some risk proteins were associated with the risk of multiple cancers, or may have a role in potential adverse outcomes. I am concerned about whether there are pleiotropy instrumental variables (cis-pQTL) in proteome-wide MR and should be considered.

Author Response: We agree with the reviewer that horizontal pleiotropy can be a concern in MR analyses. In the context of *cis* pQTL, however, horizontal pleiotropy is much less likely due to the high biological plausibility that genetic variants within or nearby a proteins' cognate gene will associate with its plasma levels (PMID: 27342221) – in our study we defined our window around protein cognate genes using recently empirically defined optimal distance to identify *cis* pQTL (PMID: 35527238). We therefore believe associations of a *cis* pQTL with protein levels and multiple traits or cancer outcomes likely represent multiple instances of vertical pleiotropy due to the presence of colocalization for any protein-trait associations reported in Supplementary Table S4. From our current understanding this implies a shared causal locus at or nearby the proteins' cognate gene between protein levels and each trait. However, we acknowledge that future work may inform other potential explanations, such as the role of intergenic variation in determining DNA structure that may impact the transcription and/or translation of large genomic loci that include both gene's leading to disease and protein's cognate gene.

3. If there are more than 3 instruments, it would be nice to see the Egger-based, median-based MRs as sensitivity analysis, and MR-PRESSO for more than 4 IVs.

Author Response: We agree that finding consistent evidence for a proteins' role in cancer risk from multiple *cis*-pQTL using methods, such as those suggested, would be favourable. However, in our study we did not find colocalised support from multiple and independent *cis* pQTL ($r^2 < 0.01$) with risk of cancer and therefore were not able to conduct these analysis. This is reported on line 277 as

“Additionally, we did not identify any proteins as a risk factor for cancer from multiple, independent cis pQTL in MR analyses”.

4. Reverse MR between cancers and risk proteins could be added for more analysis.

Author Response: We thank the reviewer for this suggestion. We have conducted a reverse MR for each protein-cancer association reported in Table 1, which can be found in Supplementary Table S2. These analyses and their results are reported in the text on Line 234:

We assessed the potential impact of cancer risk on protein levels for proteins identified in main analyses using the inverse-variance weighted and weighted median approaches and using GWAS significant [$p < 5e-08$] and independent genetic variants [clumping at $r^2 < 0.01$]. Additionally, we used the Egger intercept to assess directional pleiotropy and, where it indicated significant pleiotropy, we also conducted MR-Egger analyses. Data were harmonised as described in protein-cancer analyses. We required consistent evidence from all risk estimation methods and Bonferroni significance corrected for the number of proteins investigated to support the association of cancer risk with protein levels.

And Line 278: *“We did not find supportive evidence after correction for multiple testing for the association of cancer risk with protein levels for any protein identified in our main risk analyses (Supplementary Table S3).”*

5. F statistic could be calculated to measure the strength of each instrument.

Author Response: We have added a column in Supplementary Table S1 to indicate the F statistic and added a sentence to the main methods section in addition to the supplementary materials that indicates that we selected our cis pQTL based on F statistic > 10 on line 203:
Only cis pQTL with an F statistic > 10 were taken forward for analysis.

6. Please estimate the statistical power of the MR association, which is implemented in mRnd (<https://cnsgenomics.shinyapps.io/mRnd/>).

Author Response: Based on the median variance explain in protein levels by cis pQTL included in our analyses (1.3%) we estimate that we had at least 80% power to detect an odds ratio of between 1.11 and 1.55 for a 1 SD increase in protein levels, depending on cancer endpoint. We have now included this information in Table 1 along with a sentence on line 206:

Estimated power calculations are presented in Table 1 based on median variance explain in protein levels by cis pQTL (1.3%).

Additionally, we note that for the findings in our paper for which external validation was possible due to an external cancer risk GWAS, the majority of findings validated, which we believe lends confidence to our findings.

7. It could be more convincing to validate the findings using pQTLs data from an independent cohort.

Author Response: We agree that validation of the specific cis pQTL associations with protein levels would be a valuable exercise, in particular, across different high-throughput technologies, such as Olink and Somalogic, to assess the potential for aptamer and epitope-binding effects. Some work on a limited sample of proteins in a modest cohort has show that important differences may exist (PMIDs: 35984888, 34819519). Further work on the expected data for Olink, Somalogic, and mass-

spectrometry-based methods in the UK Biobank will provide important information on the validity of pQTLs-

8. Based on the central dogma, is there a consistency between plasma protein and blood gene?

Author Response: We thank the reviewer for this thoughtful question. While we are measuring proteins in the blood plasma, we cannot guarantee that it was expressed in the blood, and in some cases we expect that the protein is measurable in the blood having come from another tissue. At present, however, there are not sufficient data to directly estimate whether SNPs that predict a proteins' levels also predict that protein's mRNA in blood or in other tissues. This is due to there being more limited GWAS data of gene expression compared to GWAS data of protein levels, with studies of expression frequently being underpowered. However, one of us (A.M.) has recently conducted a comprehensive investigation into the overlap between eQTL and pQTL for a panel of 92 inflammatory proteins (PMID: 33067605), which suggested that the overlap may be one quarter of genetic variants that predict a protein in blood also predict its expression in at least one tissue. As further genetic studies of gene expression in multiple tissues become available it will be informative to further investigate this question.

9. More details on the clinical value of risk proteins could be added to the discussion.

Author Response: We thank the reviewers for highlighting the important questions of clinical implications from our work. We have provided three vignettes to describe possible routes to translation for protein-MR findings for cancer prevention. After further experimental and functional follow-up, as well as other pre-clinical and clinical investigations, some of these forty proteins may provide new options for prevention and be part of the strategies needed to limit the expected increase in cancer incidence. We have now added a sentence to the discussion to further comment on these potential clinical implications of our work on line 590:

After further experimental and functional follow-up, as well as other pre-clinical and clinical investigations, some of the forty proteins we have identified may provide new options for prevention and be part of the clinical strategies needed to limit the expected increase in cancer incidence.

Reviewers' comments:

Reviewer #1 (Remarks to the Author):

The authors have addressed some of my concerns, but several major concerns still exist:

1. In terms of quality control, the authors have limited their filtering to variant allele frequency (MAF). However, they overlooked several crucial factors, such as population structure and balanced case/control ratios. For instance, with respect to population structure, the authors should provide either the lambda value or the LDSC intercept value from the GWAS to ensure that population stratification issues are absent.
2. The authors should not rely on a preprint to justify the use of two distinct cutoffs. I am still concerned about the impact on the conclusion if the authors adjust these two arbitrary cutoffs.
3. While excluding trans-pQTL may diminish the impact of horizontal pleiotropy, the issue persists with cis-pQTL. Therefore, the authors should contemplate incorporating tests for horizontal pleiotropic effects to evaluate their influence on the results. Moreover, it's essential to assess the heterogeneity in the outcomes through various methods.

Reviewer #2 (Remarks to the Author):

The author has done a very good job and given a serious and detailed response to each comment. I have a remaining comment here: there is an inconsistency between the Dictionary and Supplementary Tables. Please check carefully.

The authors have addressed some of my concerns, but several major concerns still exist: **1. In terms of quality control, the authors have limited their filtering to variant allele frequency (MAF). However, they overlooked several crucial factors, such as population structure and balanced case/control ratios. For instance, with respect to population structure, the authors should provide either the lambda value or the LDSC intercept value from the GWAS to ensure that population stratification issues are absent.**

We thank the reviewer for their comments on the quality control for cis pQTL that we include in our analyses. We would add, however, that we have performed steps in addition to filtering based on MAF that are crucial for Mendelian randomisation (MR) studies. This includes filtering based on F statistic > 10 to address potential for weak instrument bias and harmonising the direction of effect and alleles for cis pQTL between protein and cancer GWAS summary statistics.

We agree that population structure is important, however, we must point out that we did not conduct these protein GWAS. Instead, we have extracted genetic instruments from multiple large, published protein GWAS' that were all conducted in populations of European ancestry where population structure was well-controlled for using best practices in the field of genetics. It is not common practice or reasonable to require a restatement of these metrics in a manuscript that uses these summary statistics over the requirement to provide clear citations, which we have done. We do not feel, therefore, that this is a valid scientific criticism or requirement of our study and it does not principally serve to undermine any of our findings.

With respect to concerns about case-control imbalance we would like to add some additional context. Reasonable criticisms of degree of case-control imbalance are aimed at the characteristics of the previously published GWAS and not to do with any analyses conducted as part of our manuscript. Of the nine cancer outcomes for the previously published GWAS' in our manuscript, for only two cancer outcomes (non-malignant melanoma and cancer of the endometrium) were there what is considered to be a modest case-control imbalance (PMID: 30104761). All other published GWAS studies that we have used do not fulfil any reasonable estimate of an imbalanced study. We would add the potential bias that one would have concern about stemming from an imbalance in cases and controls would be an increased false positive rate in the SNP associations with cancer risk in those studies. We acknowledge this general concern and, with respect to ensuring our findings are robust to this bias, we have also conducted additional MR analyses using external cancer GWAS to reduce any potential influence of false positives. We see no cause for concern given the high validation rates for our protein-cancer risk associations.

2. The authors should not rely on a preprint to justify the use of two distinct cutoffs. I am still concerned about the impact on the conclusion if the authors adjust these two arbitrary cutoffs.

We thank the reviewer for their detailed interrogation of our methods. We would like to provide greater clarity on our rationale for the choice of p-value thresholds in our study. Neither threshold we chose is arbitrary. Further, the choice of a more lenient p-value threshold for cis pQTL has precedent in a highly cited proteogenomic study published in Nature Genetics (PMID: 32895551), as well as in a preprint, both of which we referenced in our last response.

In our study, we use the conventional GWAS significance threshold of $5e-8$ as a primary threshold that is estimated to manage false positive rates in the context of a study considering the whole genome and expected LD structure. As a cis pQTL-sparing secondary threshold we used the established secondary threshold for p values in GWAS studies ($5e-5$) to identify additional cis pQTL; given that for any given protein the proportion of the genome that can be defined as cis is a fraction of one percent of the total genome length, the secondary threshold we used ($5e-5$) is still very conservative. Nonetheless, we also performed external validation where possible to further reduce the risk of false positive conclusions from our work about the role of a protein in cancer risk using either p value threshold.

We would welcome specific rebuttals to the scientific rationale of our design that we believe has allowed for the investigation of an expanded number of proteins with cancer risk in a manner that would allow for the detection of false conclusions.

3. While excluding trans-pQTL may diminish the impact of horizontal pleiotropy, the issue persists with cis-pQTL. Therefore, the authors should contemplate incorporating tests for horizontal pleiotropic effects to evaluate their influence on the results. Moreover, it's essential to assess the heterogeneity in the outcomes through various methods.

We appreciate these comments on the role of horizontal pleiotropy in MR studies and would like to provide some additional detail for how this issue should be considered in cis pQTL MR analyses.

Cis pQTL MR studies leverage the limited set of high-quality protein-associated variants that lie within a narrow window around the protein's cognate gene. However, as we outline both in our manuscript and in our responses to the reviewers, after we restricted to cis pQTL with a significant MR association and with evidence of being at a shared causal locus with cancer risk there were no proteins with more than one cis pQTL and that were independent. As such, with only one cis pQTL, other methods like those suggested by Reviewer 1 and that may regularly be applied to the analysis of more complex traits, such as BMI, were not applicable here as they require multiple independent genetic variants. The low number of genetic variants cis to the protein's cognate gene is generally considered to be outweighed by a specific and favourable characteristic of cis pQTL - their clear biological link with protein levels. This makes horizontal pleiotropy much less likely in cis pQTL MR analyses, and is a relevant feature common to any paper that investigates proteins using cis pQTL.

We would also like to acknowledge that there are limited, and very interesting, cases where we may identify protein-cancer associations for multiple proteins in a gene dense region

where it may be difficult to pinpoint which protein precisely at that locus is the causal gene. We agree with Reviewer 1 that adequate attention should be paid to potential issues around pleiotropy. We therefore value the opportunity to include text in our manuscript that outlines where reasonable caution should be paid to the interpretation of cis pQTL MR results. We have addressed this issue in the discussion section:

“Similarly, it can be difficult to interpret colocalization between a protein and cancer risk where a cognate gene sits in a genomic region with irregular haplotype structure, such as C4A associations with lung cancer on chromosome six, or when two genes that lie immediately adjacent to each other, both share similar associations with cancer risk, such as PTPN9 and SNUPN with breast cancer risk in our analyses.”

REVIEWERS' COMMENTS

Reviewer #1 (Remarks to the Author):

The novelty of the author's study lies in exploring the relationship between pQTLs and various types of cancer using MR and coloc analysis. The cited articles, such as Zheng et al., Ferkingstad et al., Folkersen et al., and Pietzner et al., have already conducted similar analyses with similar discoveries. For example, Zheng et al. identified cancer-related proteins through integrating eight cancer GWAS summary statistics with pQTLs using MR and colocalization analysis. Ferkingstad et al. also integrated the pQTL with cancer GWAS summary statistics. It is challenging to evaluate the novelty of the manuscript without a comprehensive comparison to these previous studies. Additionally, the authors did not address most of my concerns.

1. I am still not convinced by the arbitrary cutoffs defined by the authors. especially considering that the original authors in NG (PMID: 32895551) still use the canonical cutoffs (5×10^{-8}) in their method. Also, the authors only use $PP4 > 0.7$ as their colocalization cutoff, which has been frequently described as problematic due to the probability of high $PP3$, a factor not considered by the authors.
2. The authors emphasized the use of FinnGen and UKBB for external validation. However, in the replication data used by the authors, 76.92% (10/13) of the replication datasets have significant case-control imbalance issues (cases: controls $> 1:20$). For example, in Pancreatic Cancer (2352 cases / 687431 controls; cases: controls = 1:292) and Renal Cancer (3530 cases / 702568 controls; cases: controls = 1:199). Previous studies have shown that extremely unbalanced case-control ratios can generate many false signals (PMID: 30104761). In this case, the authors used GWAS results that may have a significant number of false signals for MR analysis, greatly increasing the likelihood of producing false results and conclusions.
3. For transparency and reproducibility, the authors should disclose all summary statistics and cis-pQTL data, and also share their analysis code.

Reviewer #3 (Remarks to the Author):

Smith-Byrne et al investigate 2,074 circulating proteins as potential risk factors for nine common cancers using Mendelian Randomization (MR). They employ a two-stage design, with the discovery analysis using the largest available cancer-specific GWAS and replication using GWAS summary data from the UK Biobank and/or FinnGen cohort(s). The authors perform colocalization, reverse MR, and other sensitivity analyses to interrogate the robustness of the observed cancer-protein associations and potential violations of MR assumptions. They also leverage external databases in follow-up analyses to identify potential drug targets proxied by relevant cis-pQTL instruments. Overall, this is a rigorous and informative analysis. The paper is well-written and results are appropriately presented and interpreted.

For any MR study it is important to critically examine the suitability of the GWAS summary statistics for MR analysis, so the themes raised by Reviewer 1 are relevant. However, I think the authors have responded adequately to these critiques and I don't think that the remaining issues uncover any serious methodologic flaws or cause for concern with this manuscript.

It is very clear that this study uses post-QC summary statistics, so I don't think it's reasonable to assume that only MAF filtering was applied. Also, since most of these GWAS are from peer-reviewed publications one would expect them to be of acceptable quality. However, this should not be taken for granted, so I examined a few of the references in Table 1. None of the studies seem to have QC issues. For example, looking at the studies with a large case-control imbalance, for endometrial cancer (1:8) the original GWAS publication (Ref 24) states that there was "little evidence of genomic inflation ($\lambda_{1000} = 1.004$) Using linkage disequilibrium (LD) score regression, we estimate that 93% of the observed test statistic inflation is due to polygenic signal, as opposed to population stratification." For non-melanoma skin cancer (1:15) Ref 30 reports LDSC intercept of 1.046, which does not indicate inflation due to confounding. Regarding the replication analysis, the FinnGen publication (Ref 32) clearly states that association analyses were performed using SAIGE [PMID: 30104761] which controls for case/control imbalance.

Also, just to clarify, inflation due to case/control imbalance is not something that will be readily detected by the LD score intercept test because this metric captures inflation due to population structure. Furthermore, even if both sets of GWAS summary statistics (discovery and replication) are susceptible to some type of bias and/or inflation, it's not reasonable to assume that the magnitude and direction of this bias would be similar across all these studies and that it would produce the same types of spurious signals and MR associations. Therefore, the external validation of these cancer-protein associations does strengthen the credibility of these results.

Regarding the thresholds used for colocalization ($PP.H4 > 0.70$) and cis-pQTL p-values for instrument selection ($p < 5e-05$), both are reasonable and aligned with thresholds commonly used in the literature. Since the probability of joint causality cannot be greater than the causal association with either trait, 0.70 seems like a safe choice because H_3 will be < 0.3 . From looking at the results, it seems that applying a higher threshold ($PP.H4 > 0.80$) or even ($PP.H4 > 0.90$) wouldn't have an appreciable impact of on the main findings of the study. Also, the authors use colocalization to control for LD in the region, rather than to pinpoint a single causal variant, so in the unlikely event of PP.3 (shared signal, different causal variants) being mistakenly labeled as PP.4 (shared signal, same causal variants), this wouldn't undermine the conclusions of the study.

For cis-SNPs (eQTL or pQTL) there is no reason to use the GWAS threshold of $p < 5e-08$ since association analyses are not being performed across the entire genome (however $p < 5e-08$ would be appropriate for trans-pQTLs). For example, GTEx uses FastQTL, which sets the nominal p-value that corresponds to $FDR < 0.05$ for each gene/cis-region using permutations. In practice these nominal p-values end up being

in the $p < e^{-05}/e^{-06}$ range. So perhaps the p-value threshold set by the authors uses a slightly simplistic approach, but it's not wrong. Also, if indeed the authors included weak instruments by using the $p < 5e^{-05}$ threshold, as the reviewer implies, then this would bias the MR analysis towards the null and lower the likelihood of replicating these protein-cancer pairs (not induce false positive associations).